# Viral Vector Vaccine Development and Application during the COVID-19 Pandemic

**DOI:** 10.3390/microorganisms10071450

**Published:** 2022-07-18

**Authors:** Shaofeng Deng, Hui Liang, Pin Chen, Yuwan Li, Zhaoyao Li, Shuangqi Fan, Keke Wu, Xiaowen Li, Wenxian Chen, Yuwei Qin, Lin Yi, Jinding Chen

**Affiliations:** 1Department of Microbiology, Li Ka Shing Faculty of Medicine, The University of Hong Kong, Pokfulam, Hong Kong, China; sfdeng@connect.hku.hk; 2State Key Laboratory for Emerging Infectious Diseases, The University of Hong Kong, Pokfulam, Hong Kong, China; 3College of Veterinary Medicine, South China Agricultural University, No. 483 Wushan Road, Tianhe District, Guangzhou 510642, China; hliang@scau.edu.cn (H.L.); 20191028012@stu.scau.edu.cn (Y.L.); lzhaoyao123@163.com (Z.L.); shqfan@scau.edu.cn (S.F.); wukeke@stu.scau.edu.cn (K.W.); 18306616234@163.com (X.L.); chwenxian0912@163.com (W.C.); ywqin2022@163.com (Y.Q.); 4Oriental Fortune Capital Post-Doctoral Innovation Center, Shenzhen 518055, China; pchen@ofcapital.com; 5Chinese Academy of Sciences, Shenzhen 518055, China; 6Key Laboratory of Zoonosis Prevention and Control of Guangdong Province, Guangzhou 510642, China

**Keywords:** viral vector vaccine, SARS-CoV-2, COVID-19

## Abstract

With the accumulation of mutations in SARS-CoV-2 and the continuous emergence of new variants, the importance of developing safer and effective vaccines has become more prominent in combating the COVID-19 pandemic. Both traditional and genetically engineered vaccines have contributed to the prevention and control of the pandemic. However, in recent years, the trend of vaccination research has gradually transitioned from traditional to genetically engineered vaccines, with the development of viral vector vaccines attracting increasing attention. Viral vector vaccines have several unique advantages compared to other vaccine platforms. The spread of Omicron has also made the development of intranasal viral vector vaccines more urgent, as the infection site of Omicron is more prominent in the upper respiratory tract. Therefore, the present review focuses on the development of viral vector vaccines and their application during the COVID-19 pandemic.

## 1. Introduction

Vaccines can be divided into the following categories: DNA, mRNA, inactivated, live attenuated, virion-like particle, viral vector, recombinant subunit and synthetic peptide (Figure 1). The effectiveness of vaccines is usually evaluated by animal experiments and clinical trials before the immune effect of the vaccine can be approved. To date, more than 153 candidate vaccines have entered the clinical stage, and new vaccine platforms are being enhanced. Existing COVID-19 vaccines have yielded encouraging results in both animal studies and clinical trials. As shown in Table 1, some vaccine candidates in these platforms have been approved [1].

The continuous evolution and mutation of COVID-19 not only causes serious obstacles to pandemic prevention, but also results in great challenges to the research and development of vaccines. According to the World Health Organization (WHO) statistics, five strains have now been identified as COVID-19 mutants: Alpha, Beta, Gamma, Delta and Omicron. The cross-protection of COVID-19 vaccines can be shown by the protective effect of the same vaccine on different COVID-19 variants, most of which still have protection but with less efficacy.

For viral vector vaccines, viral vectors are genetically engineered to introduce target genes encoding key antigens of pathogens. Currently, widely used viral vectors include adenovirus, measles virus, influenza viruses and poxvirus [2]. By applying proven viral vectors, it is feasible to rapidly develop vaccine candidates against emerging virus outbreaks or pandemics. Viral vectors have different entry kinetics, replication capacities and protein expression profiles, so the level of immune response and protection provided by viral vector COVID-19 vaccines may greatly vary. Viral vector vaccines combine the safety advantages of inactivated vaccines with the immunological benefits and low cost of live vaccines, which have been widely used in veterinary medicine [3]. However, the barrier to apply a viral vector vaccine in human clinical use is significantly higher. Before the COVID-19 pandemic, there were several viral vector vaccines available for various infectious disease, including Zika virus [4], HIV [5], malaria [6] and Ebola virus [7,8,9]. The urgency to develop effective vaccines during the SARS-CoV-2 pandemic has accelerated the innovative research of viral vectors.

## 2. Advantages and Challenges of Viral Vector Vaccines

The common goal of vaccines is to train the immune system to recognize the corresponding virus or certain viral protein components, thus protecting the body against infection. To achieve this goal, three major aspects have to be concerned and carefully evaluated: safety, stability and immunogenicity. Viral vector vaccines have unique advantages in these aspects. 

### 2.1. Safety

For safety concerns, viral vectors are modified by removing the virulence gene but maintaining the capacity to enter cells, which enable them to deliver target genes without causing disease. This process mimics viral infection routing. A variety of viruses have been developed as vectors, including adenovirus, influenza virus, measles virus and cowpox virus. These viral vectors do not contain disease-causing genes and even the genes that allow them to replicate have been removed in some of them. In the case of the ChAdOx1 nCoV-19 vaccine, for example, the modified chimpanzee adenovirus ChAdOx1 carries the gene of the SARS-CoV-2 spike protein into the nucleus, where it is transcribed into mRNA by DNA polymerase. mRNA then leaves the nucleus and enters the cell plasma where it binds to ribosomes to synthesize antigen proteins, which are expressed and released, followed by antigen presentation, which stimulates the body to produce an immune response (Figure 2). 

### 2.2. Stability

Some of the mRNA vaccines have to be stored at ultra-low temperatures to avoid degradation. For example, BNT162b2 from Pfizer-BioNTech needs to be stored at −80 °C to −60 °C for up to 6 months. The mRNA-1273 vaccine from Moderna should be stored at −20 °C for up to 6 months; it remains stable for up to 30 days when kept at 2 °C to 8 °C [10]. Compared to the other vaccine platforms, viral vector vaccines are more stable, requiring less stringent storage and handling conditions. Ad26.COV2.S, an adenoviral vector-based vaccine from Johnson & Johnson, can be stored and transported at −25 °C to −15 °C until the expiration date, or stored at 2 to 8 °C for 6 months [11]. This considerably increases the availability and affordability of vaccines, making vaccination more accessible worldwide. 

### 2.3. Immunogenicity

Antigen presentation and delivery are critical for the development of immunity and protection. Both humoral and cellular immunity are required for the clearance of SARS-CoV-2 according to findings from early human coronavirus research and evidence obtained following the COVID-19 outbreak [12,13,14,15]. Long-term protection without the need for several booster vaccines is possible with the antigen presentation route by the viral vector vaccine. Finally, consistent safety and high levels of immunogenicity require the long-term monitoring of adverse events after vaccination and an enhanced understanding of viral vector immunology.

#### 2.3.1. Humoral Immunity

Humoral immunity is a form of specific immunity where, upon the detection of a specific antigen, B cells create antibodies for protective purposes. Humoral immunity against viruses is achieved through vaccination or natural infection. Neutralizing antibodies are one of the most important predictors of immune protection against viral infections and are a fundamental component of humoral immunity. As different viral vectors have different entry kinetics, replication capabilities and protein expression profiles, they may elicit very different immune responses and confer very different levels of protection. The most widely used viral vector for COVID-19 vaccine development is the adenoviral vector, as adenoviruses can infect a wide range of host tissue cells [16]. Other viral vector-based COVID-19 vaccines have also undergone extensive preclinical evaluations, including vesicular stomatitis virus (VSV) [17,18,19], measles virus [20,21], oncolytic herpes simplex virus-1 (HSV-1) [22] and modified vaccinia virus Ankara (MVA) [23,24]. Although most viral vector vaccines stimulate strong humoral immunity in animal models or humans, a phase I study examining an rVSV-vectored COVID-19 vaccine candidate developed by Merck was terminated due to low immunogenicity [25].

Immunogen designs also affect viral vector vaccine-mediated humoral immunogenicity. Huang et al. discovered that immunizing animals with SARS-CoV-2 spike proteins without glycan shields resulted in stronger protective responses [26]. Based on replication-competent chimeric bovine/human parainfluenza virus type 3 (B/HPIV3) that expresses the prefusion-stabilized (S-2P) SARS-CoV-2 S spike protein, Liu et al. developed a live intranasal vector vaccine against COVID-19; in comparison to the naive variant, the prefusion-stabilized version elicited higher neutralizing antibody titers [27].

In addition, the dosage, route of immunization, homologous or heterologous primary immunization boosting methods and individual differences between vaccinators can all affect the effectiveness of the vaccine.

#### 2.3.2. Cell-Mediated Immunity

Viral vectors vaccine are capable of triggering powerful and long-lasting cellular responses, such as cytotoxic T lymphocytes, thus eradicating virus-infected cells. Antigen-specific cytotoxic T cells (CD8+) proliferate, differentiate and transform into effector T cells when triggered by antigens, and cell-mediated immunity stimulates macrophages and natural killer cells to eradicate intracellular pathogens. 

In a phase I/II clinical trial of the AZD1222 vaccine, on days 7–28 after vaccination, both CD4+ and CD8+ T cells had increased expressions of CD69 and Ki-67, as well as an increase in tumor necrosis factor (TNF) and interferon (IFN) production by CD4+ T cells; the total expression of Ki-67 by NK cells steadily increased. Individuals who got the ChAdOx1 nCoV-19 vaccine had significantly increased IFN-γ and IL-2 levels [28]. For Ad5-nCoV, the vaccine was immunogenic, prompting rapid T-cell responses in most participants. T-cell responses peaked at around 14 days following immunization, indicating a quick initiation of measurable immunological responses. In vaccine recipients, both CD4+ and CD8+ T cells were activated, especially antigen-specific CD4+ and CD8+ T cells. However, the presence of substantial pre-existing anti-Ad5 immunity reduced both specific antibody and T-cell responses elicited by the vaccination [29].

#### 2.3.3. Mucosal Immunity

The nasopharynx is the first site where COVID-19 invades the human body. However, the distribution of IgG induced by intramuscular vaccination at this site is very low, so it is difficult to resist the invasion of the virus. In addition, it was found that Omicron mainly causes upper respiratory tract infection; therefore, changing the vaccination route, combined with mucosal immunity, may be more effective in preventing infection.

The mucosal immune system can produce specific SIgA antibodies locally, forming the first line of defense against viral invasion. It is capable of eliciting a powerful immune response to external pathogenic antigens while preserving immunological tolerance to non-pathogenic antigens. Unlike most intramuscular vaccines that induce only cellular and humoral immunity, the nebulized inhalation or nasal spray vaccinations generate mucosal immunity, which prevents infection and interrupts transmission at the first point of viral invasion. Clinically, it has been found that SARS-CoV-2 Omicron variant infections exhibit a concentrated attack on the upper respiratory tract and therefore increased anti-infective effectiveness may require the maintenance of very high neutralizing antibody titers in the respiratory mucosa [30,31].

For mucosal responses, the nasal route is preferable to the parenteral route, considering its reliability and ease of application [32,33,34]. Furthermore, to prevent the replication of SARS-CoV-2 virus in the nasal epithelium, a local mucosal immune response is important and, therefore, many pharmaceutical companies and researchers have developed new modes of administration of the COVID-19 vaccine to stimulate the organism’s mucosal immune response, mainly including nasal spray administration and nebulized inhalation. The original version of the adenovirus vector vaccine developed by CanSino Biologics in China was administered by intramuscular injection, and a nebulized inhalation delivery method was later developed for clinical trials; the study is the first to report clinical data for an aerosol COVID-19 vaccine [35]. The results show that two doses of aerosol Ad5-nCoV are well tolerated, without causing any vaccine-related serious adverse events. One dose of aerosolized Ad5-nCoV, equal to a fifth of an intramuscular dose, could induce a strong cellular response, and two doses of aerosolized Ad5-nCoV can produce similar SARS-CoV-2 neutralizing antibody titers as one dose of the intramuscular vaccination. Furthermore, an aerosolized booster vaccination at 28 days after the first intramuscular injection induced strong IgG and neutralizing antibody responses. Of all the vaccines, the replicating influenza virus vector-based vaccine, DelNS1-2019-nCoV-RBD-OPT1, produced by Wantai Biopharmaceutical Company in China, which is administered as an intranasal spray, may be valuable for ease of administration and patient compliance during outbreaks [36].

#### 2.3.4. Immunity against Viral Vectors

Viral vector vaccines may develop immunity against the viral vector itself, which can be a double-edged sword. On the one hand, many viral vectors are “self-adjuvants”, which means they can stimulate the innate immune signaling cascade in response to host cells via pathogen-associated molecular patterns (PAMPs) bound to pattern-recognition receptors (PRRs), in contrast to conventional vaccines, which require adjuvants to activate innate immunity [37]. Immunity against viral vectors has also opened up new possibilities for the development of multi-disease or multi-pathogen vector vaccines, in which key protective antigens from two or more pathogens in a single vector can be used to immunize against multiple diseases, allowing for “one shot against multiple diseases” [38]. On the other hand, however, pre-existing immunity due to previous exposure and immunity to viral vectors may limit the vaccine’s effectiveness. In a phase-I trial study, individuals with significant (>1:200) pre-existing Ad5 neutralizing antibodies showed weaker antibody and T-cell responses following the injection of a recombinant Ad5-vectored COVID-19 vaccine compared to those with little (1:200) pre-existing Ad5 immunity [29]. Strategies to solve this problem include mixed vaccination. For example, the vaccine Sputnik V was administered with two different human adenovirus vectors, Ad5 and Ad26, for the first and second vaccinations, respectively, to enhance the vaccine effectiveness [39].

### 2.4. Side Effects after Viral Vector Vaccine Vaccination against SARS-CoV-2

The side effects of a vaccine are a very common occurrence; all COVID-19 vaccines currently in use have reported varying degrees of side effects. Typical side effects include short-term, mild-to-moderate injection-site pain; redness and swelling; and systemic flu-like symptoms, including fatigue, headache, muscle pain, chills, joint pain and fever.

The ChAdOx1 adenovirus vector vaccine is one of the most widely used COVID-19 virus vector vaccines available. A study published in *N. Engl. J. Med.* showed that 7 to 10 days after receiving the first dose of the ChAdOx1 vaccine, some vaccinees developed venous thrombosis and thrombocytopenia, referred to as vaccine-induced immune thrombocytopenia [40]. Meanwhile, cases of venous thrombosis, including cerebral venous sinus thrombosis (CVST) after ChAdOx1 vaccination, have been reported in several European countries. In early March 2021, 30 venous thromboembolic events were reported to the European Medicines Agency (EMA) among the approximately 5 million people who received the ChAdOx1 vaccine at that time [41]. Statistics from a survey showed that 62 vascular cerebrovascular adverse events with a close temporal association to the COVID-19 vaccination were identified in Germany as of 14 April 2021, of which 45 cases were CVST and 11 patients died. Statistics found that patients receiving the first dose of the ChAdOx1 vaccine had a 10-fold to 90-fold higher incidence of CVST than the normal population, and a 10-fold higher risk of CVST after ChAdOx1 vaccination compared to mRNA-based vaccines. Additionally, CVST with severe thrombocytopenia has been reported within two weeks after vaccination with Ad26.COV2.S [42,43]. The underlying mechanism of action of these thrombotic events after vaccination with the adenoviral vector-based SARS-CoV-2 vaccine remains incompletely understood, although various data and hypotheses have been proposed. The possible association with the adenoviral vector vaccine encoding the SARS-CoV-2 spike protein suggests that the mechanism of action is dependent on, or at least includes, the adenoviral vector used in this vaccine [44].

In conclusion, the incidence of rare thrombotic and thrombocytopenic events was higher after adenoviral vector-based anti-SARS-CoV-2 vaccines compared to mRNA-based anti-SARS-CoV-2 vaccines.

## 3. The Application of Viral Vector Vaccines during the COVID-19 Pandemic

Viral vector vaccines have emerged as one of the leading candidates for developing an effective, safe and mass-producible vaccine against the ongoing COVID-19 pandemic [45]. MVA, adenovirus, para-influenza virus, Sendai virus, rabies, Newcastle disease virus and influenza viruses were used as vectors to develop the vaccine [46]. Viral vectors have different entry kinetics, replication capacities and protein expression profiles, so the level of immune response and protection provided by viral vector COVID-19 vaccines may greatly vary. 

As of 15 May 2022, 25 virus vector-based vaccination candidates (21 non-replicating and 4 replicating) were included in clinical trials, according to the WHO draft outlook for COVID-19 vaccine candidates. Table 2 summarizes some of the viral vector vaccines used against COVID-19 that are in late-stage development, which can be divided into non-replicating and replicating vaccines [1]. 

### 3.1. Non-Replicating Viral Vector Vaccine

The most widely used non-replicating viral vector for COVID-19 vaccines is the adenovirus vector. Adenovirus is a double-stranded non-enveloped DNA virus with more than 300 different serotypes of adenovirus, which can target a wide array of parallel tissues for cell infection, opening up an important way to design new vaccines against COVID-19 and other pathogens [47]. Human adenoviruses AdHu5 [29,48,49], AdHu26 [50,51,52] and chimpanzee adenovirus [53,54] are now being employed as viral vector vaccines against SARS-CoV-2 in clinical trials. 

Ad26.COV2.S is an intramuscular viral vector vaccine produced by Janssen (Johnson & Johnson) with the SARS-CoV-2 spike protein gene inserted into a non-replicating human adenovirus type 26 (Ad26) vector. The vaccine has been found to protect hamsters and rhesus macaques against SARS-CoV-2 in preclinical trials [55]. The vaccine’s clinical trials began in June 2020, with a phase III trial involving approximately 43,000 people [11]. A single dose of Ad26.COV2.S proved effective against severe-critical disease, including hospitalization and death, as well as symptomatic COVID-19 and asymptomatic SARS-CoV-2 infection [11,56,57]. On 27 February 2021, the US Food and Drug Administration (FDA) granted the vaccine emergency use authorization and conditional marketing authorization from EMA on 11 March 2021. However, recently, on 6 May 2022, the FDA announced that it had restricted the authorized use of Janssen’s COVID-19 vaccine to certain people over the age of 18 years who were clinically unable to receive other approved COVID-19 vaccines or who were only willing to receive Janssen’s vaccine. Janssen was restricted because of the risk of blood clots caused by the vaccine. FDA determined that Janssen’s COVID-19 vaccine caused thrombosis with thrombocytopenia syndrome after the latest analysis, evaluation and investigation of the reported cases. The potential risks posed by the vaccine to other vaccinees have exceeded the known and potential benefits of its preventive fee deduction endurance, and that the authorized use of the vaccine needs to be restricted [43]. 

Another approved adenovirus vector vaccine is AZD1222, also known as ChAdOx1 nCoV-19, a non-replicating vaccine with a viral vector designed by Oxford University in collaboration with AstraZeneca, using a modified non-replicating chimpanzee adenovirus vector (ChAdOx1) as the vector for encoding the full-length codon-optimized S protein of SARS-CoV-2 [58]. It was placed on the WHO emergency use list on 15 February 2021 and was the first vaccine in the WHO global vaccine distribution system. The AstraZeneca vaccine is currently used in 173 countries and territories and has an average efficacy of 70.4% and a maximum of 90% in phase-III clinical testing covering 24,000 people [59]. Although less effective than the mRNA vaccine from Pfizer and Modena, the AZD1222 vaccine can be stored in a refrigerator at 2 °C to 8 °C for 6 months [60], which has the advantage of easy storage and transportation, and is inexpensive, making it easy to promote mass vaccination in developing countries. Although a paper in the New England Journal of Medicine noted that, for AZD1222 vaccine recipients under the age of 50 years, symptoms of blood clots occurred at least once in about every 50,000 vaccine recipients, current evidence suggests that the benefits of this vaccination still outweigh the risks [61]. 

Sputnik V is a non-replicating adenovirus vector-based vaccine developed by the Gamaleya Institute of Epidemiology and Microbiology [39]. The vaccine uses two different human adenovirus vectors, Ad5 for prime and Ad26 for boost, to enhance the vaccine’s effectiveness. This allows the body to produce a more effective immune response. In contrast, if the same vector is used, the immune system triggers specific immunity against vectors and may reduce the effectiveness of the vaccines. The phase-I-and-II trial results were published in The Lancet [62]. According to the paper, all participants induced high SARS-CoV-2 antibodies with no serious adverse events detected; most adverse reactions were mild. The phase-III trial data demonstrated that the vaccine was 91.6% effective based on its ability to prevent symptomatic infections [63].

The single-dose viral vector vaccine AD5-nCOV (Convidecia) for COVID-19 was jointly developed by the Chinese Academy of Military Sciences and CanSino Biologics. In February 2021, global data from the phase-III trial showed that the vaccine was 65.7% effective in preventing moderate symptoms of COVID-19 and 91% effective in preventing severe disease [64]; it is similar to other viral vector vaccines, such as AZD1222, Gam-COVID-Vac and Ad26.COV2.S. Additionally, a randomized phase-IV trial suggested that heterologous boosting with Convidecia following initial vaccination with CoronaVac was safe and more immunogenic than homologous boosting [65]. Recently, a phase-I study published in The Lancet evaluated the safety, tolerability and immunogenicity of nebulized Ad5-nCoV administered by nebulized inhalation, which showed that the neutralizing antibody response to the two-dose nasal spray version of Convidecia was similar to that of the existing one-dose injectable version [35]. In February 2021, it was approved by the China Food and Drug Administration (CFDA) for conditional marketing. On 19 May 2022, China’s adenovirus vector COVID-19 vaccine was included in the emergency use list, recommended for the vaccination of people over 18 years of age. In addition to the traditional intramuscular injection, the vaccine can also be administered by inhalation, which provides a new means for the prevention and control of COVID-19. The single-dose regimen and normal refrigerator storage requirements (2° to 8 °C) of AD5-nCOV could make it a favorable vaccine choice for many countries. Studies have shown that, after vaccination with two doses of the inactivated virus, the enhancement with the AD5-nCOV vaccine was safe and the neutralizing antibody level was significantly better than that of homologous immunization. 

### 3.2. Replicating Virus Vector Vaccine

Replicating viral vector vaccines can produce many virus particles upon entry into cells, and the number of viral genes they carry increases as the virus vector replicates, leading to higher protein expression of target antigens, thereby inducing a strong and sustained immune response against SARS-CoV-2. Compared to non-replicating viral vector vaccines, its main potential problem is that the protein expression of viral vector itself may induce toxic effects.

BriLife, also known as IIBR-100, is a recombinant VSV viral vector COVID-19 vaccine candidate with a replication capacity [18]; it was developed by the Israel Institute for Biological Research (IIBR). The mechanism involves replacing the glycoprotein (G) gene of VSV (vesicular stomatitis virus) with the spike protein of the SARS-CoV-2 virus. IIBR completed clinical trials and commercialized the vaccine in collaboration with NRX Pharmaceuticals in the USA. A study on hamsters showed that a single dose of the vaccine was safe and effective in protecting against COVID-19 [66]. The vaccine has been studied in a phase-IIb/III clinical trial (NCT04990466); subjects received two intramuscular doses of the vaccine (prime boost) over a 28-day interval. Clinical data indicate that IIBR-100 is an effective and protective vaccine against SARS-CoV-2 infection and shows no signs of safety concerns.

DelNS1-2019-nCoV-RBD-OPT is a COVID-19 vaccine candidate developed by the Wantai Biopharmaceutical Company in collaboration with Xiamen University and the University of Hong Kong [36]. It is an influenza viral-vectored nasal spray COVID-19 vaccine that induces dual antibodies against the influenza virus and SARS-CoV-2, which shows its unique advantage during influenza seasons. The surface protein of the influenza virus vector is modifiable to adapt to circulating influenza virus strains, which is a good solution to the problem of reduced vaccine efficacy due to “anti-vector immunity” when booster immunization is administered. Furthermore, as the vaccine is administered by nasal spray, it can induce mucosal immunity in the upper respiratory tract. In preclinical studies, when hamsters were challenged with COVID-19 one day after a single dose of the vaccine or nine months after the booster, lung pathology was significantly reduced and the hamsters did not lose weight, indicating that the vaccine provided long-lasting protection for at least nine months. The researchers also found that the virus had a broad spectrum of effects on Omicron and other variants of concern. In addition, it could provide cross-protection against H1N1 and H5N1. A phase-II clinical trial was conducted in China; the aim of this study was to evaluate the immunogenicity and safety of the vaccine for intranasal spraying according to different immunization regimens. In addition, the effect of pre-existing antibodies to influenza virus on the immunogenicity of the vaccine was also investigated [67]. The results of the study showed that the vaccine was well tolerated in adults, and weak T-cell immunity as well as weak humoral and mucosal immune responses against SARS-CoV-2 were detected in the peripheral blood of vaccine recipients. Further studies are needed to validate the safety and efficacy of intranasal vaccines as a potential addition to the current intramuscular SARS-CoV-2 vaccine pool. To date, the vaccine is undergoing large-scale phase-III clinical trials in the Philippines, South Africa, Vietnam and Colombia to directly verify its protective effect against COVID-19 [68]. In conclusion, the preclinical study elucidated the immunologic mechanism of the nasal spray vaccine, and the preliminary phase I and II clinical results showed that the nasal spray vaccine was well tolerated and could activate multiple immune responses. Although antibody seroconversion is weak, ongoing phase-III trials will further confirm the efficacy of the nasal spray vaccine. Considering the complementary immune protection mechanism of nasal spray and spinal vaccines, the nasal spray vaccine will become an important supplement to the current COVID-19 vaccine.

COH04S1 is a viral vector vaccine based on the modified vaccinia virus Ankara (MVA) for the prevention of COVID-19 infection, which co-expresses spike and nucleocapsid antigens from SARS-CoV-2, developed by the City of Hope Medical Center [69]. A phase-I study was conducted on 189 volunteers, to determine the optimal dose of the COH04S1 vaccine and to assess its safety [70]. A phase-II trial was also initiated to determine the immune response to the COH04S1 vaccine compared to the EUA SARS-CoV-2 vaccine [71].

## 4. Conclusions and Prospects

Currently, vaccines remain the most effective tool in fighting the COVID-19 pandemic. However, the researchers found that various vaccines, such as BNT162b2 and Ad5-nCoV, had varying degrees of reduced protection against variant strains [72,73]. Therefore, it is of great significance to develop more effective vaccines and change the vaccination strategy to control infections and alleviate the pressure on public health. Worldwide, clinical trials have highlighted the unique advantages of viral vector vaccines compared with other candidates. The genetic engineering of both the viral vector and inserted immunogen allows continuous optimization and rapid renewal against coming variants. The main administration routes of vaccines include intramuscular, oral and intranasal vaccinations. Omicron mainly causes upper respiratory tract infections while intramuscular vaccines often target lower respiratory tract infections [74]. Therefore, it is of great significance to develop novel, highly effective intranasal vaccines. Furthermore, replicating viral vectors mimic natural infections, induce cytokines and other stimulatory molecules and provide an effective adjuvant effect, increasing innate immunity as well as humoral, cellular and mucosal immune responses. 

Ultimately, the research on and development of COVID-19 still faces considerable challenges and requires continuous collaboration. Despite the clinical success of viral vector vaccines, pre-existing immunity to carrier particles is still a major obstacle to achieving effective immunogenicity. Rare adverse events are another challenge for viral vector vaccines. Additionally, the long-term monitoring of the safety and immunogenicity after vaccination is needed to obtain a better understanding of viral vectors, making it easier for further optimization. 

Looking ahead, we believe that viral vector vaccines not only play a vital role in combating the COVID-19 pandemic, but also providing a rapid and effective vaccine platform. The current dominant strain of COVID-19 also makes the development of the intranasal viral vector vaccine more urgent, as the infection site of it is more prominent in the upper respiratory tract. Furthermore, the development of a viral vector-based multi-target or universal vaccine may improve the efficiency of vaccines and help us cope with future outbreaks. 

## Figures and Tables

**Figure 1 microorganisms-10-01450-f001:**
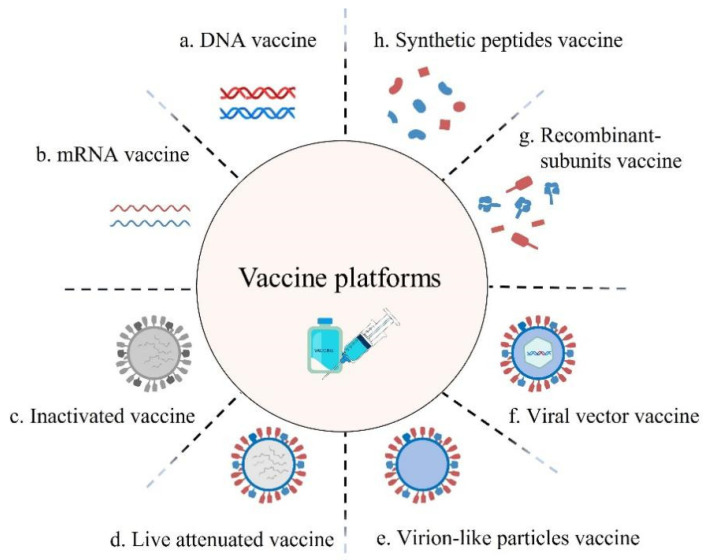
Different platforms used for development of vaccines.

**Figure 2 microorganisms-10-01450-f002:**
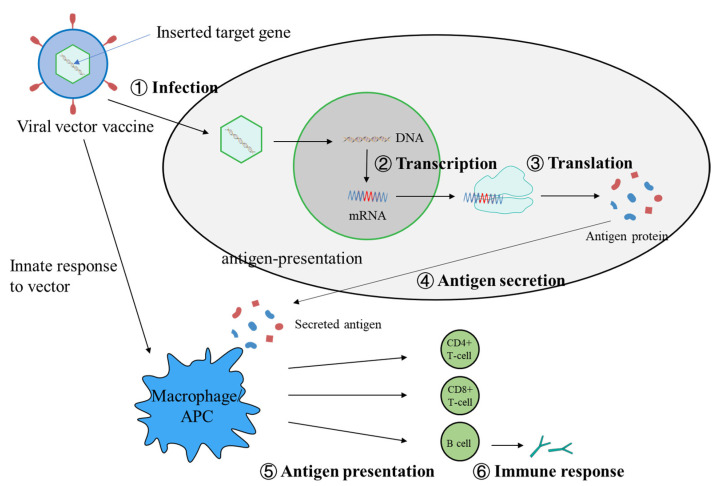
Mechanism of viral vector vaccines induce specific cellular and antibody responses.

**Table 1 microorganisms-10-01450-t001:** Landscape of vaccine candidates for COVID-19.

Candidate	Platform	Developer	Approved Date
BNT162b2	mRNA vaccine	Pfizer BioNTech	31 December 2020
ChAdOx1-S (AZD1222)	Viral vector vaccine	Oxford/AstraZeneca	16 February 2021
Ad26.COV 2.S	Viral vector vaccine	Johnson & Johnson	12 March 2021
mRNA-1273	mRNA vaccine	Moderna	30 April 2021
BBIBP-CorV	Inactivated vaccine	Sinopharm	7 May 2021
CoronaVac	Inactivated vaccine	Sinovac	1 June 2021
BBV152 COVAXIN	Inactivated vaccine	Bharat Biotech	3 November 2021
NVX-CoV2373	Recombinant subunit vaccine	Novavax	17 December 2021

**Table 2 microorganisms-10-01450-t002:** Landscape of viral vector candidate vaccines in late clinical development for COVID-19.

Vaccine Candidate	Viral Vector	Platform	Administration Route	Developers	Clinical Trials
ChAdOx1-S (AZD1222)	Chimpanzee adenovirus	VVnr	IM	Oxford/AstraZeneca	Phase 4
Convidecia(Ad5-nCoV)	Adenovirus type 5	VVnr	IM/IH	CanSino/Chinese Academy of Military Medical Sciences	Phase 4
Sputnik V (rAd26-S+rAd5-S)	Adenovirus 26 and adenovirus 5	VVnr	IM	Gamaleya Research Institute/Health Ministry of the Russian Federation	Phase 3
Ad26.COV2.S	Adenovirus 26	VVnr	IM	Johnson & Johnson	Phase 4
GRAd-COV2	Gorilla Adenovirus	VVnr	IM	ReiThera Srl/Lazzaro Spallanzani National Institute for Infectious Diseases	Phases 2/3
DelNS1-2019-nCoV-RBD-OPT1	Influenza virus	VVr	IN	University of Hong Kong/Xiamen University/Beijing Wantai	Phase 3
IIBR-100 (rVSV-SARS-CoV-2-S)	Vesicular stomatitis virus	VVr	IM	Israel Institute for Biological Research	Phases 2/3
BBV154	Chimpanzee adenovirus	VVnr	IN	Bharat Biotech	Phase 3
NDV-HXP-S	Newcastle Disease virus	VVr	IN/IM	Sean Liu, Icahn School of Medicine at Mount Sinai	Phases 2/3

VVnr: viral vector (non-replicating); VVr: viral vector (replicating); IM: intramuscular; IN: intranasal; IH: inhaled.

## Data Availability

Not applicable.

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
