# Peer review of "Viral Vector Vaccine Development and Application during the COVID-19 Pandemic"

_microorganisms, 2022, doi:10.3390/microorganisms10071450_

Round 1
Reviewer 1 Report
The manuscript provides a succinct listing and overview of the manufacturers, vaccine efficacy and usability perspectives of the licensed viral-vectored vaccines for covid-19.
I have only a minor point regarding the readability of the manuscript. The use of long sentences impairs readability eg.
- the second sentence in the abstract
- the first sentence in the introduction.
- the first sentence of section 2.3.2
I suggest a profound revision of the manuscript to address this point, which will improve the quality of the manuscript.
Author Response
Response: Thank you for your kind advice. We have made corresponding modifications to the manuscript according to your suggestions.
Reviewer 2 Report
The authors have presented this review as information on importance of viral vector vaccine development in COVID19 pandemic. This review requires significant changes in multiple ways:
a) Several sentences are floppy and non-scientific, some sentences are incomplete or not well thought out. Too many to mentioned please go carefully and instead of writing long sentences and covering information in one statement, consider splitting long sentences.
b) While I am not rejecting the paper, considering significantly improved content would be added to the next draft, this manuscript does not add any value on Viral vector vaccine development than what is already covered in recent reviews on the same topic. (https://www.mdpi.com/1999-4915/14/4/759; https://www.ncbi.nlm.nih.gov/pmc/articles/PMC8805485/pdf/1124-9390_29_3_2021_328-338.pdf)
c) Content is not up to date and key publications related to this topic are not cited. Please perform a thorough literature search before making further changes to this manuscript.
d) Figures are not high informative. Also, advantages/disadvantages of viral vector based vaccine compared to all the mentioned vaccines need to be highlighted/discussed.
e) As the title says "application in the COVID-19 pandemic", please discuss the currently existing vaccine known so far (both approved or in consideration, individual or in combination with other options) and then format on how viral vector based vaccines weighs over other vaccines (such as mRNA vaccine, inactivated vaccine etc.) in some ways or how vaccine is better. The title would be better justified with it.
f) The information about vaccine efficacy against different variant is missing and should be discussed.
g) A major part discusses about the immunogenicity (including different immune responses) that does not require to be a separate topic since all vaccines work in a similar way (including mRNA vaccines etc.).
h) Confirm the details mentioned in Tables. For example, approval dates are not correct Moderna and Astra Zenca vaccines were approved in January 2021.
i) The authors superficially touch a vaccine in one statement and switches to the next vaccine in subsequent statement without concluding the former one. Therefore, the reason for transition is not clear. For example, Line 155 to Line 164.
j) Conclusion section is not informative and require reformatting
Author Response
Response: Thank you for the suggestions. Here are the point-by-point response to the reviewer’s comments.
a) We have revised the sentences of the article according to the reviewers' suggestions.
b) Thank you for suggestions. The two review articles mentioned by the reviewers and our manuscript are all about viral vector vaccines for COVID-19, but with different focuses. The first article focuses on the history of viral vector vaccines and their application in COVID-19, and the different routes of administration; the second article is a detailed account of each COVID19 viral vector vaccine. Our article, on the other hand, summarizes the advantages and disadvantages of viral vector vaccines in terms of immunity, complementing but not conflicting with the two articles mentioned by the reviewer.
c) We have made changes based on the reviewers' suggestions.
d)&e) We have discussed the advantages and disadvantages of viral vector vaccines compared with other type vaccines in the article, including the better stability of viral vector vaccines, the longer periods of time to preserve and simpler preservation conditions than mRNA vaccines, and the side effects of viral vector vaccine.
f)The vaccine efficacy against different variants has been added and discussed. According to WHO statistics, five strains have now been identified as COVID-19 mutants: Alpha, Beta, Gamma, Delta and Omicron. The researchers found that various vaccines such as BNT162b2 and Ad5-nCoV, had varying degrees of reduced protection against variant strains.
g) We disagree with this comment of the reviewer. Because viral vector vaccines do differ from other types of vaccines in terms of immunity. For example, the virus vector vaccine mentioned in the article causes an immune response against the virus vector itself, and this kind of immune response is very important because on the one hand it may have some side effects on the effectiveness of the virus vector vaccine, and on the other hand, the immune response against the virus vector can also be used to prevent the virus vector itself (e.g., the influenza virus vector based COVID-19 vaccine can prevent both SARS-COV-2 and influenza virus.). In addition, viral vector vaccines, because they have more diverse immunization routes, some viral vector vaccines can induce mucosal immunity by nasal spray or inhalation, for example, which is an immune response not found in other vaccines.
h) We have reconfirmed the information in the table and there are no errors. The approved date r in the table refers to the World Health Organization (WHO) granted emergency use listing, not the FDA emergency use.
i) All the examples we mention in the article are complementary to the content of the individual chapters. For example, line 155 to line 164, mentioned by the reviewer, all of these different viral vector vaccines were administered by nasal spray or inhalation, to increase mucosal immunity, this is in line with the theme of the chapter.
j) The conclusion section has been reformatted.
Reviewer 3 Report
The review „Viral vector vaccine development and application in the 2 COVID-19 pandemic“ discussed the development of recombinant viral vector vaccines and their application during the COVID-19 pandemic. The authors discussed in particular the safety, stability, and immunogenicity of the developed COVID-19 vaccines. Although the topic is very interesting, however, it is very primitive and requires a major revision.
- What about the real efficacy of these vaccines under clinical conditions? Please discuss also the side effects of SARS-CoV-2 vaccines in particular the severe side effects such as lymphadenopathy and TTS and their mechanisms. Also please discuss the causes of vaccine hesitancy and authors' recommendations. Authors may cite the following reviews/papers that might help to address the required points (DOI: 10.1111/ane.13451, https://doi.org/10.51585/gjm.2021.2.0006, DOI: 10.1016/S1473-3099(21)00224-3, doi: 10.1016/j.intimp.2021.107970, doi: 10.1111/imm.13443, doi: 10.3390/ijms221910791, doi: 10.1016/j.ijid.2021.08.013
Minor
- Please write the full name for all abbreviations that mentioned for the first time in the manuscript
- Line 29: Synthetic should be synthetic
- Line 83: Figure 2. Please explain the figure in more detail to stand alone.
- Line 206: covid-19 should be COVID-19
- Line 216: Please delete (WHO)
- Line 237: The should be the
- References: 10, 17, 27, 40, 43, 44, 51, 56, 59, 60, 71 should be „Initials small“
Author Response
Response: Thank you for the suggestions. We have revised the article based on reviewers' minor suggestions. And we also add an chapter discuss the side effects after viral vector vaccine vaccination in chapter 2.4.
Round 2
Reviewer 2 Report
The authors have made necessary amendments and provided adequate response to most of the points raised by me in my previous review. Therefore, I endorse the manuscript for publication.
Author Response
Thanks to the reviewer for your kind help and advice.
Reviewer 3 Report
The manuscript is now improved and I recommend its publication after minor revision. Several paragraphs require adding reference/s. Also please revise the references?

Author Response
Thanks to the reviewers for the advice and help, we had revised the references.